# Performance Evaluation of Artificial Recharge–Water Intake System Using 3D Numerical Modeling

**Jae-Young Lee *** and **Tae-Young Woo**

Fusion Research Institute, Sinwoo Engineering Co., Ltd., Seoul 06184, Korea; lovewty@naver.com
*   Correspondence: vennard386@gmail.com; Tel.: +82-2-6959-3038

**Abstract:** In this study, 3D detailed numerical modeling was performed to evaluate the performance of an artificial recharge–water intake system installed to secure agricultural water in drought areas. Using a 3D irregular finite element grid, a conceptual model was constructed that reflected the actual scale of the study area and artificial recharge–water intake system and considered the characteristics of saturated–unsaturated aquifers. The optimal design factors for the artificial recharge system were derived through the constructed conceptual model, and were reflected to evaluate the individual performance of the artificial recharge and water intake system in the study area. Finally, an optimal operating scenario for the artificial recharge and water intake system was developed. The operation scenarios were composed of an appropriate injection rate and water withdrawal for each period from March, when the demand for agricultural water was low, to June, when the dry season and farming season overlapped, considering the target water withdrawal amount (30,000 tons) of the region, derived from water budget analysis. The proposed results are expected to be very useful in the future for the efficient operation and management of artificial recharge–water intake systems installed in drought areas.

**Keywords:** artificial recharge; water intake; detailed numerical modeling; performance evaluation; operation scenario

## 1. Introduction

Due to recurrent drought, it is not easy to secure water resources, so there are many areas that are experiencing difficulties in cultivating crops. South Korea has implemented policies and technologies focusing on surface water resources such as dams and reservoirs, but surface water resources sensitive to climate change have limitations as a solution to drought, and they require long-term preparation. Conversely, groundwater resources have relatively little evaporative loss and are not sensitive to climate change, so they can complement the limitations of surface water resources [1]. Accordingly, the paradigm of domestic water resources policy is changing with the development and expansion of alternative water resources employing groundwater to stably secure high-quality water in the event of an extreme drought. However, groundwater resources are also decreasing due to a decrease in natural recharge, excess extraction, and increased risk of contamination. Therefore, in order for groundwater to become a more effective water resource to respond to climate change, it is necessary to develop artificial recharge technology that can artificially increase high quality groundwater resources [2]. The artificial groundwater recharge technique is a useful method to continuously secure water resources by artificially injecting water into an aquifer by installing an injection well. Compared to surface dams, there are certain hydrological and economic advantages, so the use of the artificial recharge method is increasing for both long-term and short-term underground storage [3].

The artificial groundwater recharge technique was proposed as an effective method of securing water resources in preparation for uncertain future climate change [4], and the range of artificial recharge and appropriate artificial recharge techniques were reviewed

considering the characteristics of aquifers [5]. An artificial recharge evaluation model through 3D groundwater modeling was developed for Hancheon basin in Jeju Island using MODFLOW [6], and the optimal location of the artificial recharge well was evaluated by comparing the pumping rate change in the existing pumping well according to the location of the injection well using the groundwater model [7]. In addition, MODFLOW has been used to analyze the groundwater flow in a ditch filled with homogeneous anisotropic soil [8], and a numerical analysis model was developed to examine the application of groundwater artificial recharge in the Yongding River basin in Beijing, China [9]. Meanwhile, an underground drainage system that can mitigate landslide damage was developed and constructed on site; the stability of landslides and slopes was reviewed using a three-dimensional finite element program (Plaxis 3D) [10].

In relation to artificial groundwater recharge, various studies have been conducted internationally, but there are very few case studies based on numerical analysis in which actual water shortage areas are set, the recharge potential is evaluated in consideration of the target water withdrawal in the watershed, and where the operation conditions of the recharge rate and water withdrawal by period are presented in detail. Therefore, in this study, the optimal design factors for artificial recharge and water intake facilities were derived by using detailed numerical modeling for the area where actual artificial recharge–water intake system construction has been completed. The performance of each artificial recharge system and water intake system was then evaluated, and an optimal operation scenario for supplying agricultural water suitable for the region during the drought period was presented.

## 2. Materials and Methods

### 2.1. Study Area

The study area is located in Ungok-ri, Hongsung-gun, in the Chungnam province, and consists of agricultural land for paddy and field farming (Figure 1). The geography of the basin is surrounded by mountains with a height of about 350 m, and the large and small tributaries originating from these mountains join together to form two major streams (Shingok stream and Ungok stream). The two streams have a stream width of about 15 m, a channel width of about 2 m, and an ordinary water depth of about 0.1 m; the range of streamflow variability is highly dependent on rainfall variability. During the busy farming season (April to June), there is relatively little rainfall, so there is almost no streamflow. As continuous groundwater extraction and depletion of groundwater are occurring rapidly, artificial groundwater recharge technology is being considered. As a result of analyzing the precipitation status through meteorological data (Seosan meteorological station, 2012~2021) for 10 years in the target area, the average annual precipitation was 1106.8 mm/year. In addition, about 40% of the total annual precipitation was concentrated between July and August, and about 20% occurred between April and June, the farming season, indicating a small amount of precipitation. The main geological status of the study area is mainly composed of granitic gneiss, flaky granite, and migmatite belts of Precambrian. On the east side, biotite granite of the Cretaceous penetrates metamorphic rocks, and on the west side, sedimentary rocks from the Muryangri Formation, presumed to be Permian, are distributed. In addition, the alluvial layer is very developed due to the hilly topography. Although the average annual groundwater level in the study area fluctuates slightly by period, it is distributed at about 2 m below the ground surface. The groundwater level repeatedly rises or falls depending on whether or not there is rainfall, and accordingly, the water level drops to about 5 m below the surface during the dry season and farming season, from April to June.

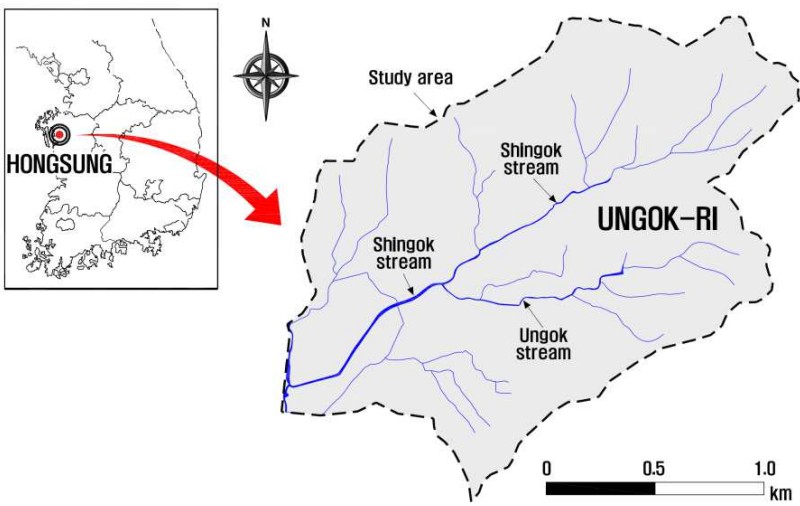

**Figure 1.** Location map of the study area.

*2.2. Methods*

For detailed numerical modeling of the artificial recharge–water intake system, a 3D, FEM-based SEEFLOW3D [11] (Cha et al., 2017) model was adopted and an irregular finite element grid was applied to implement artificial injection and water intake facilities more precisely [11]. In consideration of the physical properties, it was attempted to accurately reflect the behavior of groundwater flow in the saturated–unsaturated aquifer. In order to build a conceptual model using Digital Elevation Model (DEM) data of the digital topographic map, a modeling range of 2450 m × 2350 m was built within the boundary of the mountain ridge, which is a watershed, and the stratum information of the drilling column obtained through on-site drilling survey data was utilized. Three numerical layers were composed of an alluvial layer, weathered soil, and bedrock, and input data for the stratum characteristics of the numerical model were composed using the results of field hydraulic tests and grain size analysis in the unsaturated zone (see Table 1). A mixed artificial injection system (1 ditch + 4 vertical wells), an intake system (intake well, horizontal well), and three observation wells near the downstream part of the study area for monitoring groundwater level fluctuations were reflected in the model. A computational mesh with a total of 24,650 nodes and 43,596 elements was built by constructing irregular three-dimensional elements. As shown in Figure 2, the outside of the target area was set as the no-flux boundary, and the rivers within the modeling area were set as the time-series river stage (Dirichlet boundary). In addition, in order to derive the optimal design factors for the artificial injection system to develop an operational scenario for an artificial recharge–water intake system, three major design factors were set and evaluated as follows:

1.  Optimal size of media in the ditch of the artificial recharge facilities from various media sizes (10/30/50/70/90 mm)
2.  Optimal spacing of vertical wells for artificial recharge facilities from various vertical well intervals (5/10/15 m)
3.  Optimal screen opening ratio of horizontal wells for water intake facilities from various opening ratios (10/20/30%)

**Table 1.** Distribution of Geological Layer and Hydraulic Conductivity.

| Media | Layer | Depth (GL.–m) | | Hydraulic Conductivity (cm/s) | |
| --- | --- | --- | --- | --- | --- |
| | | Range | Average | Range | Average |
| Alluvium | 1 | 1.5~5.8 | 3.1 | $4.05 \times 10^{-5} \sim 1.14 \times 10^{-3}$ | $4.11 \times 10^{-4}$ |
| Weathered soil | 2 | 1.5~6.5 | 3.7 | $5.82 \times 10^{-5} \sim 4.10 \times 10^{-3}$ | $7.00 \times 10^{-4}$ |
| Bedrock | 3 | 5.8~12.3 | 8.7 | $1.40 \times 10^{-6} \sim 2.13 \times 10^{-4}$ | $7.20 \times 10^{-5}$ |

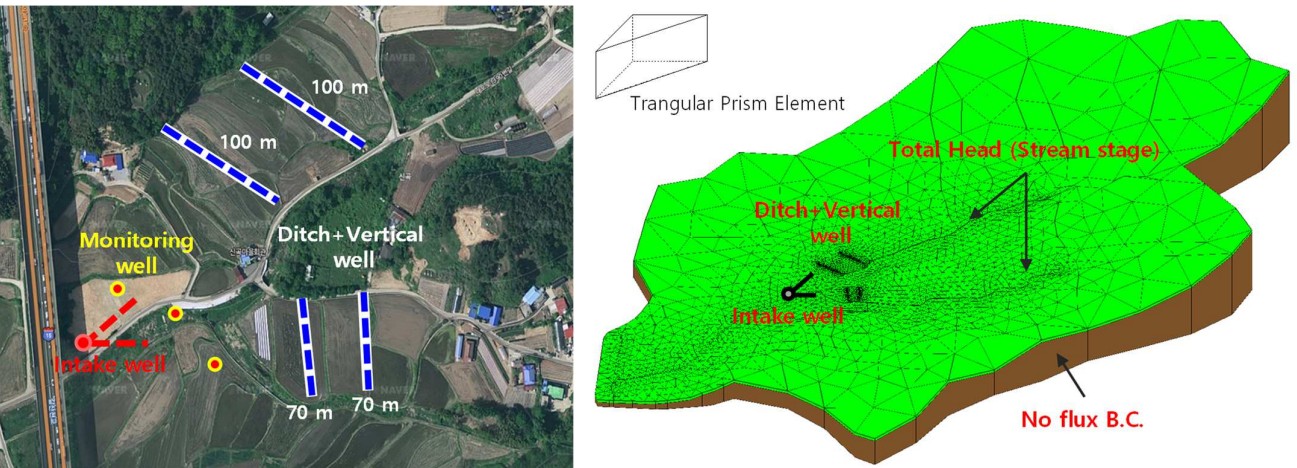

**Figure 2.** Configuration of conceptual model reflecting the recharge facilities and intake well in the target area.

Based on the optimal design factors, the performance of each artificial injection and intake system in this study area was evaluated, and finally, an optimal operation scenario of the artificial recharge–water intake system was developed.

## 3. Results

### 3.1. Evaluation of Optimal Size of Media in the Ditch

Because the ditch of the artificial recharge facility is located 1.0 m below the ground surface, it is necessary to prevent the upper soil from flowing into the artificial recharge facility as much as possible. A filter media is required because the water flow in the ditch should be stable and uniform. Therefore, in order to derive the optimal size of the media in the ditch of the artificial recharge facility, numerical simulations were performed for each media size of 10, 30, 50, 70, and 90 mm, and the optimal size of the media was derived by setting the time when the groundwater level was full in the ditch. The hydraulic conductivity and porosity were used as modeling input data for each media size. The hydraulic conductivity was applied as 35 m/day for 10 mm, 1000 m/day for 30 mm, 3500 m/day for 50 mm, 6000 m/day for 70 mm, and 8600 m/day for 90 mm, and the porosity was used, ranging from 0.34 for 10mm to 0.28 for 90mm, respectively [12]. For the artificial injection conditions according to the media size, the injection location was designated at regular intervals on the upper element of the ditch, and the inflow flux boundary condition was applied at an injection rate of 45 ton/day. The numerical simulations were performed for a total of 1 month, with a computational time step of 1 h, the time for filling the ditch for each media size was estimated, and the amount of penetration of water injected through the ditch into the bottom of the ditch was compared.

As a result of the simulations, the time to fill the groundwater level for each media size was 16 h at 10 mm, 10 h at 30 mm, 5 h at 50 mm, 4 h at 70 mm, and 3 h at 90 mm (Figure 3a). The infiltration rate was 38 ton/day at 10 mm, 41 ton/day at 30 mm, 45 ton/day at 50 mm, 47 ton/day at 70 mm, and 45 ton/day at 90 mm (Figure 3b). In addition, Figure 4 shows the spatial distribution of the groundwater level by time in the ditch according to the artificial injection. Thus, the fill time is fastest at 90 mm and the maximum rate of infiltration is shown at 70 mm, but 50 mm is the optimal size of media considering the economic feasibility because the fill time and infiltration rate are similar at a media size of more than 50 mm.

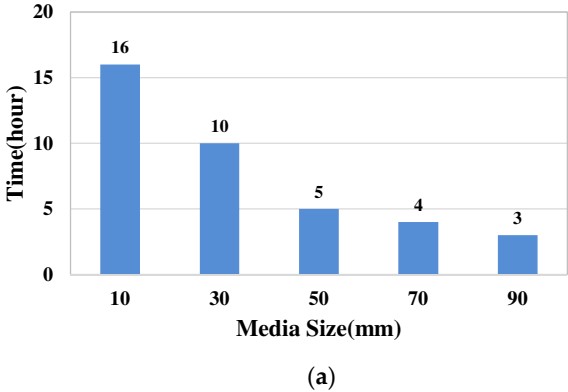

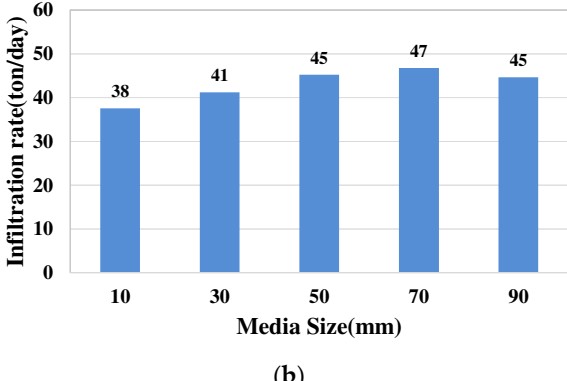

(**a**)　　　　　　　　　　　　　　　　　　(**b**)

**Figure 3.** Time for filling the ditch and infiltration rate according to media size: (**a**) Time for filling the ditch; (**b**) Infiltration rate.

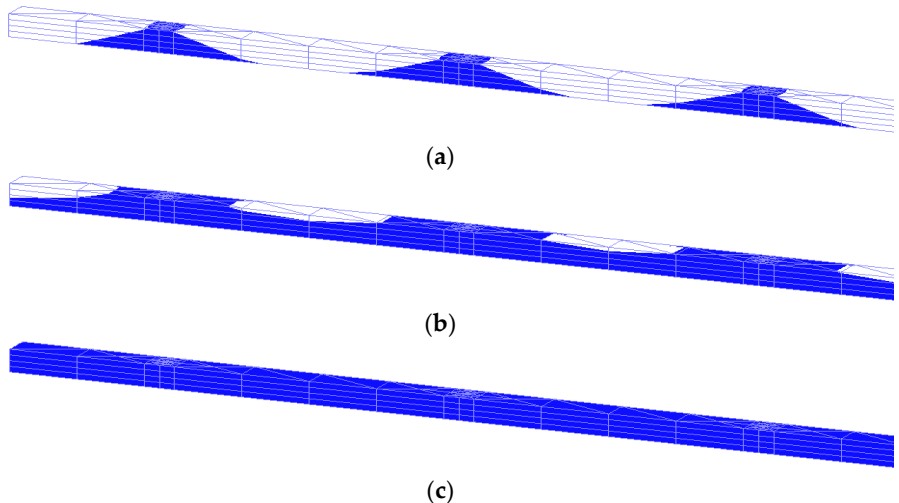

**Figure 4.** Spatial and temporal variation of the groundwater level in the ditch according to artificial recharge (media size 50 mm): (**a**) 1 h; (**b**) 3 h; (**c**) 5 h.

### 3.2. Evaluation of Proper Spacing of Vertical Wells in the Ditch

In an artificial injection system, vertical wells are very effective because they can be directly introduced into the lower strata. When multiple vertical wells are installed, a large amount of artificial recharge is possible. However, because the cost of construction and maintenance increases with the number of wells, it is necessary to determine their appropriate spacing and number. Therefore, in order to derive the proper spacing of vertical wells for artificial recharge facilities, the vertical well spacing was divided into 5, 10, and 15 m in mixed structures (ditch + vertical well), and the efficiency for each interval (ratio of aquifer infiltration to injection rate) was calculated after setting the target groundwater level rise of less than 0.1 m below the ground surface, in consideration of the depth of crop growth (Figure 5).

First, a numerical simulation of the steady state was performed to predict the spatial distribution of the groundwater level in the modeling area. As a result of the steady state simulation, the groundwater level distribution in the target watershed was EL. 21.23~318.42 m, and it was simulated with a distribution similar to the topographic elevation in the watershed (Figure 6).

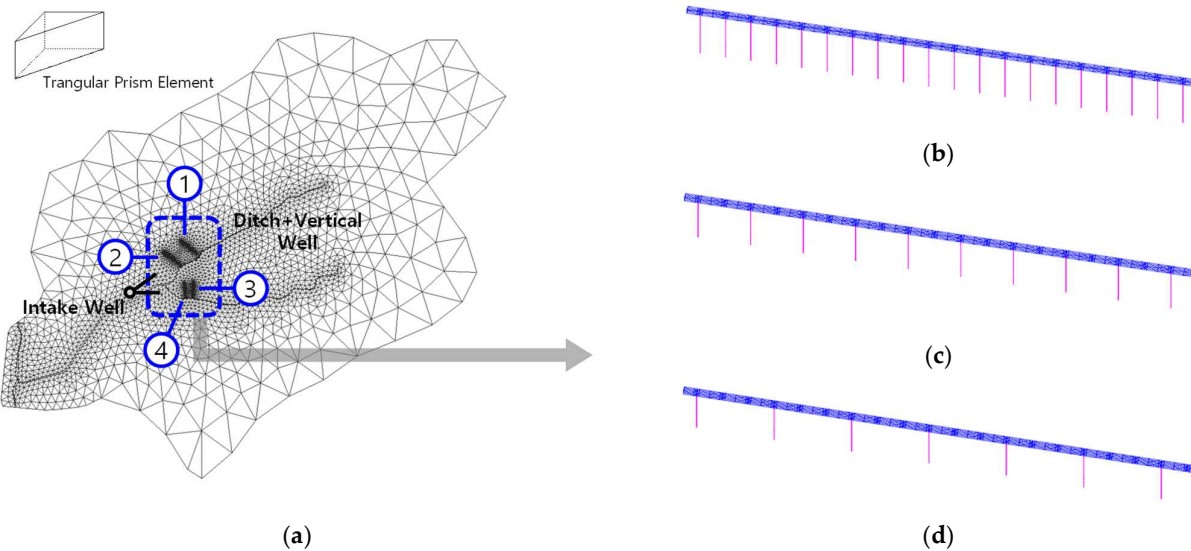

**Figure 5.** Configuration of 3D conceptual model of vertical well spacing: (**a**) Mesh generation reflecting the recharge facilities and Intake well. (**b**) Vertical well interval: 5 m. (**c**) Vertical well interval: 10 m. (**d**) Vertical well interval: 15 m.

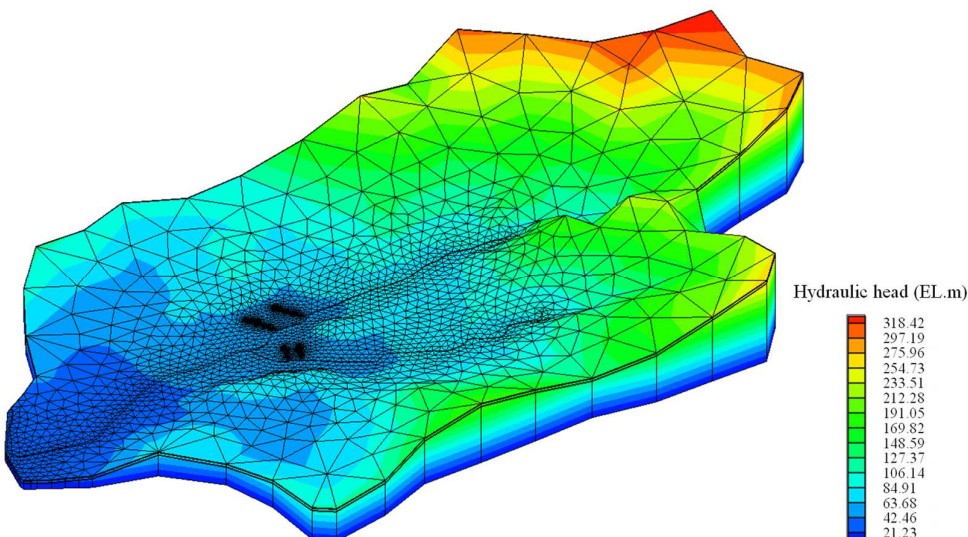

**Figure 6.** Steady-state simulation result of study domain.

For the artificial injection conditions, the injection rate for each vertical well and the ditch injection rate according to the vertical well spacing calculated from the injection test data by applying the inflow boundary condition to the vertical well and the element at the top of the ditch were used (Table 2) [13]. As a result of the simulation, at an interval of 5 m, the aquifer penetration for the injection rate of 224.8 ton/day was simulated as 184.0 ton/day, indicating an efficiency of 81.9%. At an interval of 10 m, the aquifer penetration for the injection rate of 209.2 ton/day was simulated as 185.2 ton/day, indicating a high efficiency of about 88.5%. On the other hand, intervals of 15 m, the aquifer penetration for the injection rate of 199.6 ton/day was simulated as 158.3 ton/day, which showed the lowest efficiency of 79.3% (Table 3, Figure 7). Figure 8 shows the time-series behavior of the groundwater level within the ditch, including vertical well spacing of 10 m after artificial injection, and all four ditches show that the groundwater level rises gradually after the start of injection and the fill time is similar. In this way, the injection efficiency was simulated, and in consideration of the economic feasibility and maintenance aspects of the vertical well construction, the proper spacing of the vertical wells was determined to be 10 m.

**Table 2.** Dimensions of artificial recharge facilities and injection rate conditions.

| Vertical Well Interval (m) | Vertical Well Number | | Vertical Well Injection Rate (ton/day) | | | | | Ditch Injection Rate (ton/day) | | | | | Total Injection Rate (ton/day) |
|---|---|---|---|---|---|---|---|---|---|---|---|---|---|
| | 100m Ditch (①,②) | 70m Ditch (③,④) | Ditch ① | Ditch ② | Ditch ③ | Ditch ④ | Sum | Ditch ① | Ditch ② | Ditch ③ | Ditch ④ | Sum | |
| 5 | 20 | 14 | | 1.29 | | | 87.67 | 42.7 | 36.1 | 35.9 | 22.5 | 137.2 | 224.8 |
| 10 | 10 | 7 | | 2.06 | | | 70.14 | 42.7 | 36.9 | 37.1 | 22.5 | 139.1 | 209.2 |
| 15 | 7 | 5 | | 2.51 | | | 60.29 | 42.7 | 36.9 | 37.1 | 22.5 | 139.1 | 199.6 |

**Table 3.** Injection rate and infiltration rate by vertical well interval.

| Division | | Injection (ton/day) | | | Infiltration (ton/day) | | |
|---|---|---|---|---|---|---|---|
| | | Vertical Well Interval | | | Vertical Well Interval | | |
| | | 5 m | 10 m | 15 m | 5 m | 10 m | 15 m |
| Ditch | ① | 42.7 | 42.7 | 42.7 | 34.2 | 34.9 | 35.1 |
| | ② | 36.1 | 36.9 | 36.9 | 32.2 | 33.2 | 33.2 |
| | ③ | 35.9 | 37.1 | 37.1 | 32.4 | 33.4 | 33.5 |
| | ④ | 22.5 | 22.5 | 22.5 | 20.1 | 21.2 | 21.2 |
| | subtotal | 137.2 | 139.2 | 139.2 | 118.8 | 122.7 | 123.0 |
| Vertical Well | ① | 25.8 | 20.6 | 17.6 | 20.1 | 19.0 | 10.6 |
| | ② | 25.8 | 20.6 | 17.6 | 18.8 | 18.1 | 10.0 |
| | ③ | 18.0 | 14.4 | 12.6 | 13.1 | 12.5 | 7.3 |
| | ④ | 18.0 | 14.4 | 12.6 | 13.1 | 13.0 | 7.4 |
| | subtotal | 87.6 | 70.0 | 60.4 | 65.2 | 62.6 | 35.3 |
| Total | | 224.8 | 209.2 | 199.6 | 184.0 | 185.2 | 158.3 |

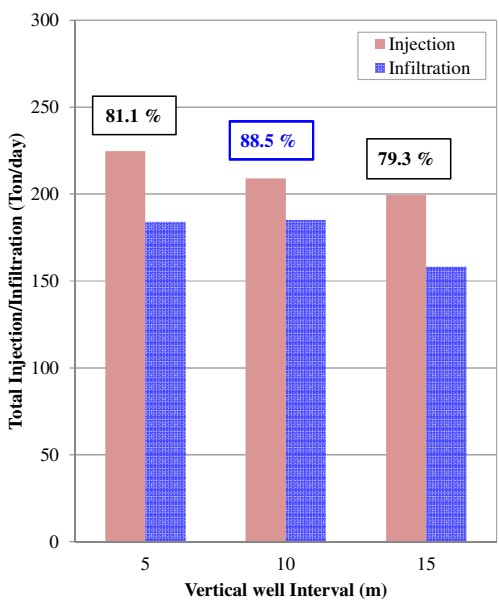

**Figure 7.** Efficiency (total injection rate/infiltration rate) by vertical well interval.

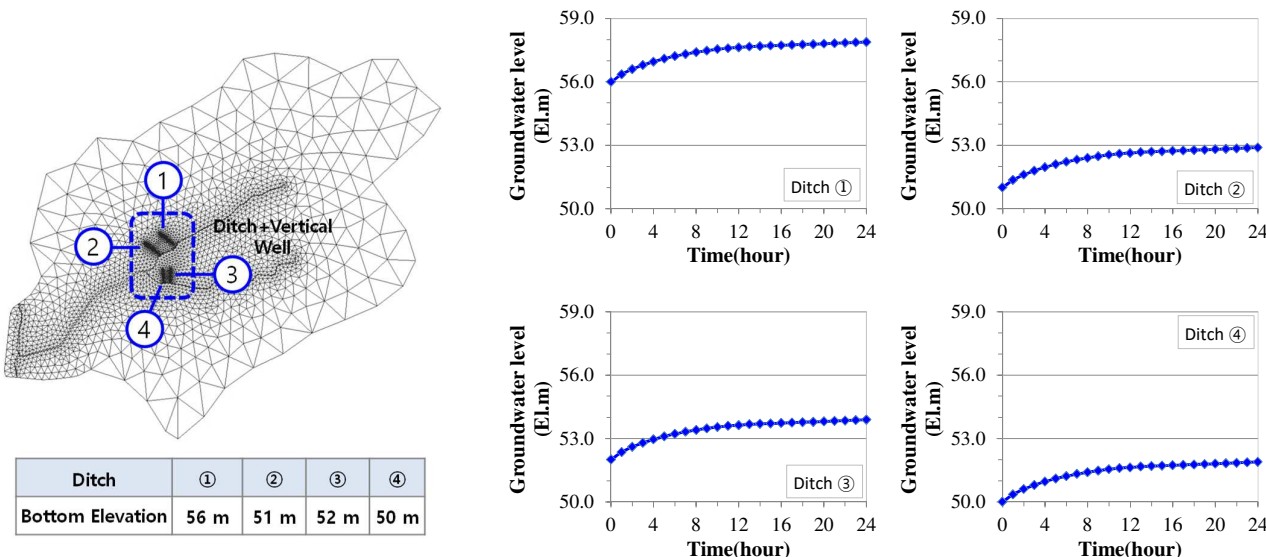

**Figure 8.** Changes of groundwater level in ditch ①, ②, ③, and ④.

### 3.3. Evaluation of Optimal Opening Ratio of Horizontal Well

The intake method for artificially recharged groundwater is divided into vertical type and horizontal radial type. The vertical water intake method takes water from the intake well configured only in the vertical direction, and maintenance and economic feasibility are low. On the other hand, the horizontal radial water intake method installs a well in the aquifer vertically and connects several horizontal wells to the side wall to take in a large amount of water. Although this method requires specialized technology, it is easy to maintain, and thus economic feasibility is high. Therefore, in this section, in order to derive the optimal opening ratio, which is the main design factor of the horizontal radial water intake method, the opening ratio was divided into 10, 20, and 30% and compared with the water level changes at the three downstream observation wells. The simulated conditions were set in such a way that after artificial recharge at 210 ton/day for 30 days, water intake was performed through a horizontal well at 500 ton/day for 10 days.

In this conceptual model, a horizontal well mesh is configured inside the model area reflecting the characteristics of the aquifer, and the inflow of groundwater into the horizontal well from the aquifer can be computed by entering the dimensions of the horizontal well (diameter, length, and opening ratio). The governing equation for this can be described as follows:

$$q = A_o \times k \times (h - H) \tag{1}$$

where $q$ is the rate of groundwater inflow into the horizontal well [$L^3 t^{-1}$], $A_o$ is the effective pore area [$L^2$], $k$ is the hydraulic conductivity of the horizontal well screen [$LT^{-1}$], $h$ is the groundwater level in the aquifer [L], and $H$ indicates the water level in the horizontal well [L]. The effective pore area of Equation (1) is defined as follows (Delleur, 2007) [14]:

$$A_o = [2 \times 3.14 \times \left( \frac{D}{2} \right) \times L] \times O_a \times 50\% \tag{2}$$

where $D$ is the horizontal well diameter [L], $L$ is the horizontal well length [L], and $O_\alpha$ is the horizontal well opening ratio [%].

On the other hand, the loss due to friction in the flow in the horizontal well can be expressed by the Darcy–Weisbach equation, as in Equation (3).

$$\frac{\Delta h_f}{L} = \frac{f V^2}{2 g D} \tag{3}$$

where $\Delta h_f$ is the friction head loss in the horizontal well [L], $f$ is the friction factor, $V$ is the fluid velocity in the horizontal well [LT$^{-1}$], and $g$ is gravitational acceleration (9.81 m/s$^2$). Additionally, $f$ is calculated as a function of the Reynold's number and can be expressed in the form of Haaland's equation, as follows:

$$f = \left[ 1.8 \log_{10}\left( \frac{6.9}{R_e} \right) + \left( \frac{\varepsilon}{3.7D} \right)^{1.1} \right]^{-2} \quad (4)$$

where $R_e$ is the Reynolds number and $\varepsilon$ is the horizontal roughness coefficient [L].

Therefore, the hydraulic conductivity of the horizontal well screen can be obtained by combining Equations (3) and (4) (Birch, 2004):

$$k = \frac{2gD}{V} \left[ 1.8 \log_{10}\left( \frac{6.9\nu}{VD} + \left( \frac{\varepsilon}{3.7D} \right)^{1.11} \right) \right]^{2} \quad (5)$$

where $\nu$ is the kinematic viscosity of water [L$^2$T$^{-1}$].

As a result of the simulation, the groundwater level fluctuations at the three observation wells dropped 0.10, 0.19, and 0.20 m at 10, 20, and 30% of the opening ratio, respectively (Figure 9). The results reproduced the general behavior of the groundwater level rising due to artificial recharge and falling due to water intake In addition, as the water intake capacity increased as the opening ratio increased, the groundwater level drop increased. However, when the opening ratio was more than 20%, the fluctuation range of the groundwater level was small. Consequently, it was predicted that there would be no significant change in the amount of water intake at an opening ratio of more than 20%, and the appropriate opening ratio of the horizontal well was evaluated to be 20%.

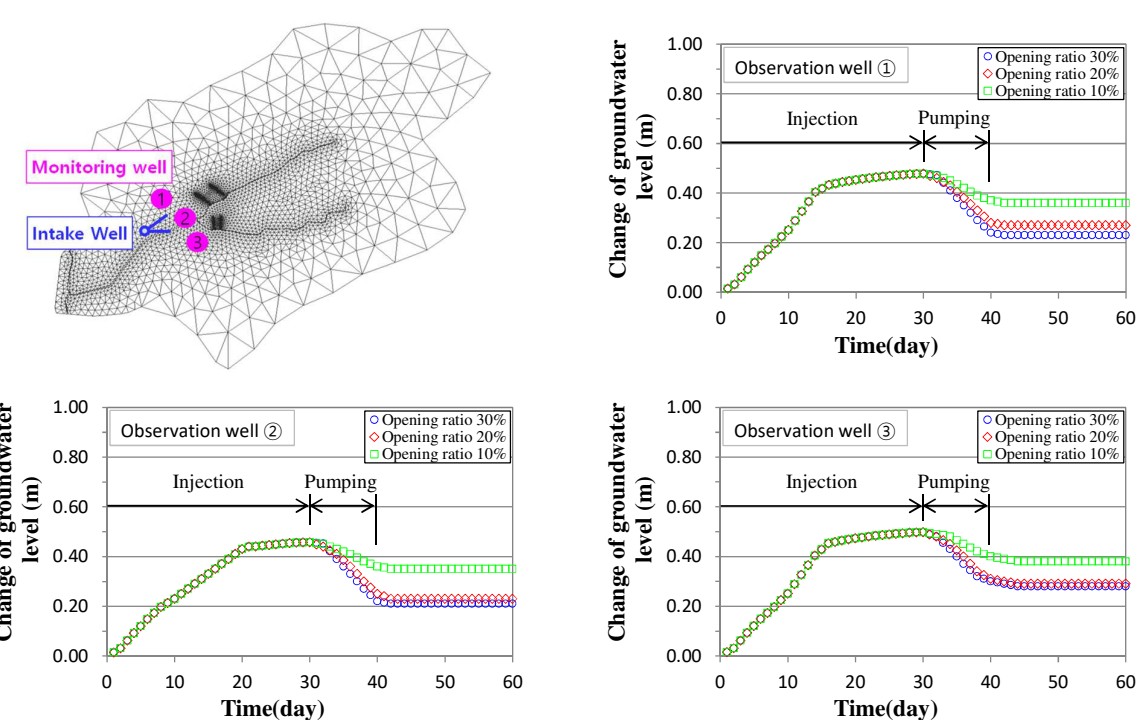

**Figure 9.** Temporal changes of groundwater level according to screen opening ratio (observation well ①, ② and ③).

### 3.4. Model Calibration

To verify the numerical model, the measured value of the groundwater level change of the observation well following the injection of the artificial recharge facility constructed on the field and the numerical modeling results were compared and calibrated. In the actual study area, five observation wells for injection experiments and groundwater level

monitoring were installed at intervals of 15 m only on Line No.1 of the four ditch-vertical well lines, which were 5 m apart from the line. The structure of the ditch-vertical well of Line No.1 was a mixed type of ditch of 50 mm media size and a vertical well of 10 m intervals installed in consideration of the design factors derived through the optimal evaluation of the artificial recharge facility described above (see Figure 10). In this way, after reflecting on the current status of Line No. 1 installed on the site in the model, the injection conditions of the site and modeling were set equal to injecting 200 tons into the ditch for 54 h and injecting 45 tons into the vertical well, and then the groundwater level changes were compared. The rise of groundwater level at the site was monitored as 49.0 cm in Well No.1, 42.5 cm in Well No.2, 38.9 cm in Well No.3, 40.3 cm in Well No.4, and 44.6 cm in Well No.5, resulting in an average increase of 43.1 cm. On the other hand, the groundwater level fluctuation predicted by the modeling was simulated as an average increase of 44.3 cm, with a rise of 45.8 cm in Well No.1, a rise of 45.2 cm in Well No.2, a rise of 44.0 cm in Well No.3, a rise of 44.3 cm in Well No.4, and a rise of 42.2 cm in Well No.5 (see Table 4). Therefore, the root mean square error (RMSE) of the actual groundwater level fluctuation measurement and the modeling result was 3.6%, indicating that the prediction accuracy of the numerical simulation was good.

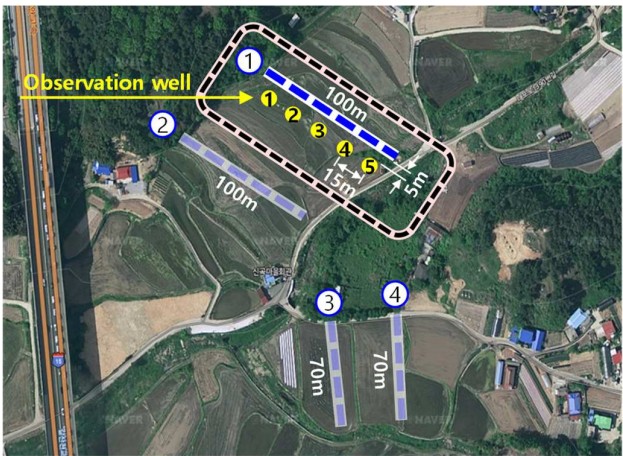

**Figure 10.** Location of field injection test and observation wells.

**Table 4.** Comparison of groundwater level rise by measurement and modeling.

| Groundwater Elevation (cm) / Observation Number | 1 | 2 | 3 | 4 | 5 | Average |
|---|---|---|---|---|---|---|
| Measurement | 49.0 | 42.5 | 38.9 | 40.3 | 44.6 | 43.1 |
| Modeling | 45.8 | 45.2 | 44.0 | 44.3 | 42.2 | 44.3 |

*3.5. Performnance Evaluation of Artifical Recharge–Water Intake System*

The conceptual model built for performance evaluation of the artificial recharge–water intake system was used to predict the fluctuations in the downstream according to the injection and water intake conditions, reflecting the optimal design factor values of artificial recharge—water intake facilities derived through 3D detailed numerical modeling. First, the spatial distribution of the groundwater level in the modeling area was simulated through a steady-state flow analysis, and a transient simulation was then performed. The change in the groundwater level in the downstream region was compared to the modeling conditions, and the average value of the three observation wells was used. In order to find a suitable artificial injection condition for this area, the injection conditions were configured in three cases of 100, 150, and 200 ton/day through four mixed-type artificial recharge facilities, and the total simulation time and injection period were set to 1 month, and the computational time step was 1 h. As a result, the artificial injection was simulated to

rise by 0.29, 0.40, and 0.48 m at 100, 150, and 210 ton/day, respectively (Figure 11a). The rise of groundwater level was similar in the case of artificial injection rates of more than 210 ton/day, so it was estimated that the injection of 210 ton/day for 1 month was the most appropriate for the artificial recharge performance of this study area. In order to evaluate the water intake capacity of the water intake system after the artificial injection is completed, as a result of comparing the change in groundwater level by configuring the water intake conditions in three cases of 300, 400, and 500 ton/day and setting the withdrawal period to 1 month, it was simulated as a fall of 0.16, 0.18, and 0.24 m at 300, 400, and 500 ton/day, respectively (Figure 11b). Thus, the drop in groundwater level at a water withdrawal of 500 ton/day does not affect the storage capacity and initial groundwater level in this study area, and the water supply efficiency within a short period is the highest, so it is estimated to be an appropriate water withdrawal amount.

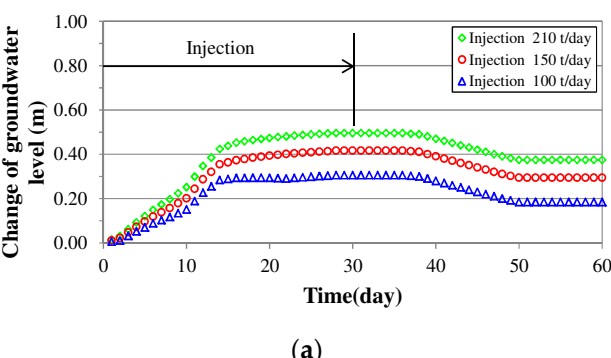
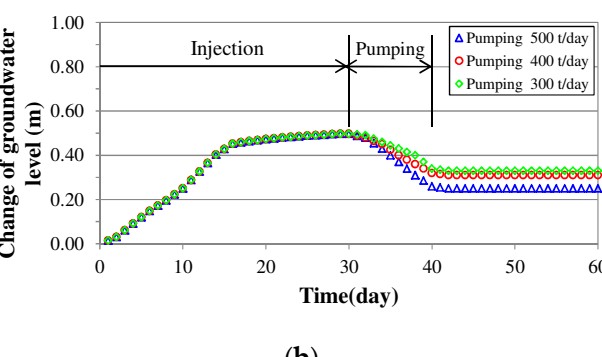

(**a**)                                                                 (**b**)

**Figure 11.** Time-series groundwater level behavior by amount of recharge and withdrawal. (**a**) water injection (**b**) evaluation the water intake capacity.

### 3.6. Optimal Operation Scenario for the Artificial Recharge–Water Intake System

In order to derive an optimal operation scenario for a sustainable water supply in the event of a drought in this study area, we referred to the agricultural water demand analysis data and the water budget analysis results [15]. In the target area, a shortage of about 30,000 tons of supply compared to demand occurs during the period from April to June, when the dry season and the farming season overlap. Therefore, an operating system was implemented that artificially recharges in March when the demand is relatively low, and water intake was carried out from April to June, and this was reviewed through detailed 3D numerical modeling. The operating stages and scenarios of the artificial recharge–water intake system for the modeling conditions were composed as follows (Table 5):

- Stage 1: Injection rate of 210 ton/day for 30 days (1 March~31 March)
- Stage 2: Injection rate of 400 ton/day for 60 days and withdrawal rate of 500 ton/day for 60 days (1 April~31 May)
- Stage 3: Shut down of recharge and water intake operation (1 June~)

**Table 5.** Optimal operation scenario for artificial recharge and water intake facilities.

| Step | Injection Period | Injection Rate (m³/day) | Total Injection Rate (ton) | Pumping Rate (m³/day) | Total Pumping Rate (ton) |
|------|------------------|-------------------------|----------------------------|-----------------------|--------------------------|
| ① | 1 March~31 March | 210 | 6300 | 0 | 0 |
| ② | 1 April~31 May | 400 | 30,300 | 500 | 30,000 |
| ③ | 1 June~ | Shut down | | | |

As a result of simulating the groundwater level change at the three observation wells in this operation stage, as a total of 6300 tons were injected for 30 days at a 210 ton/day injection rate in the first operation stage, the groundwater level rose by about 0.48 m. In the second stage of operation, following the first stage, a total of 24,000 tons were injected

at 400 ton/day for 60 days, resulting in a cumulative injection amount of 30,300 tons. At the same time, as a total of 30,000 tons were withdrawn for 60 days at a 500 ton/day withdrawal rate, the groundwater level dropped by about 0.08 m from 0.48 m to 0.40 m. In the third stage of operation, about 30,000 tons of the required water supply for this study area was withdrawn in the second stage of operation, so all operations for recharge and water intake were stopped, and it was simulated that groundwater level gradually decreased and then stabilized (see Figure 12).

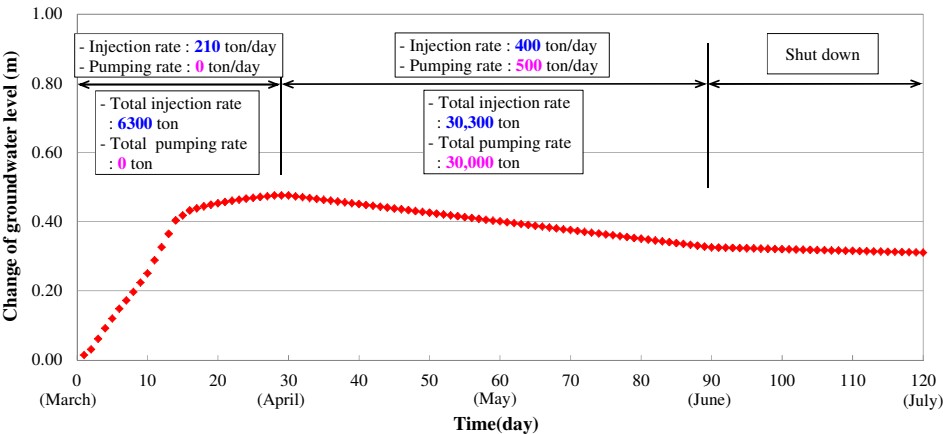

**Figure 12.** Time-series groundwater level behavior for artificial recharge and water intake.

## 4. Conclusions

Using 3D detailed numerical modeling, the optimal design factors for an artificial recharge and water intake facility were derived for drought areas where the actual artificial recharge–water intake system construction was completed. The optimal operating scenario for supplying agricultural water to the research area was then suggested through performance evaluation. The results obtained from simulation are summarized as follows:

(1)  In order to derive the optimal media size within the ditch of the artificial recharge facility, the media size was divided into 10, 30, 50, 70, and 90 mm. As a result of modeling by setting the fill time of the groundwater level in the ditch, 50 mm was determined as the optimal media size in consideration of economic feasibility because the fill time and infiltration rate were similar at more than 50 mm.

(2)  The vertical well intervals were divided into 5, 10, and 15 m to derive the appropriate vertical well interval for the artificial recharge facility, and the efficiency for each interval was estimated after setting the target groundwater level rise at less than 0.1 m below the ground surface in consideration of the depth of crop growth. As a result, efficiency was 81.9, 88.5, and 79.3% at intervals of 5, 10, and 15 m, respectively, which showed that the highest efficiency and the proper spacing of the vertical wells was determined to be 10 m in consideration of economic feasibility and maintenance according to the actual vertical well construction.

(3)  In order to derive the proper opening ratio of the horizontal well, which is the main design factor of the water intake facility, the opening ratio was divided into 10, 20, and 30%, and comparative evaluation was performed based on the average value of groundwater level fluctuations at three observation wells. As a results, the groundwater level dropped 0.10, 0.19, and 0.20 m at 10, 20, and 30% of the opening ratio, respectively. At an opening ratio of more than 20%, the fluctuation of the groundwater level was small, and it was predicted that there would be no significant change in the water intake capacity, and the appropriate opening ratio of the horizontal well was determined to be 20%.

(4)  A conceptual model was constructed by reflecting the optimal design factor values derived for the performance evaluation of the artificial recharge–water intake system, and the groundwater level fluctuations were predicted according to the in-

jection conditions. After dividing the artificial injection conditions into 100, 150, and 210 ton/day, comparative evaluation was performed through the average value of groundwater level fluctuations at three observation wells. As a result, it was simulated that the groundwater level rose by 0.29, 0.40, and 0.48 m, respectively, and the increase in groundwater level was similar in the case of artificial injection of more than 210 ton/day. Therefore, it was estimated that it was most appropriate to inject 210 ton/day for 1 month for the artificial recharge performance of this study area. In addition, in order to evaluate the water intake capacity of the water intake system after the artificial injection is completed, the water intake conditions were divided into 300, 400, and 500 ton/day, and the average value of the change in groundwater level at the three observation wells was used for comparative evaluation. As a result, it was simulated that the groundwater level dropped by 0.16, 0.18, and 0.24 m, respectively, and the drop in the groundwater level at 500 ton/day water intake does not affect the groundwater storage capacity and the initial groundwater level in this study area. Moreover, because the water supply efficiency within a short period was the highest, it was estimated that water intake at 500 ton/day was the most appropriate.

(5) Based on the optimal design factor values and performance evaluation results of the artificial recharge–water intake facility, the operation scenario of the artificial recharge–water intake system for sustainable water supply was configured by referring to the agricultural water demand analysis data and water budget analysis results in this study area. As a result of simulation, a stabilized groundwater level was confirmed through numerical modeling.

The results of this study can be utilized for the efficient operation and management of artificial recharge–water intake systems installed in drought areas in the future. It is expected that the 3D detailed numerical modeling method used in this study can be very usefully utilized to quantitatively evaluate the performance and effect of artificial recharge–water intake systems.

**Author Contributions:** Conceptualization, methodology, writing—review and editing, and supervision, J.-Y.L.; software, validation, formal analysis, investigation, data curation, writing—original draft preparation, and visualization, T.-Y.W. All authors have read and agreed to the published version of the manuscript.

**Funding:** This research was supported by the Korea Environment Industry and Technology Institute (KEITI) through the Demand Responsive Water Supply Service Program, funded by the Korean Ministry of the Environment (MOE), grant number No. 2018002650001.

**Institutional Review Board Statement:** Not applicable.

**Informed Consent Statement:** Not applicable.

**Data Availability Statement:** Not applicable.

**Conflicts of Interest:** The authors declare no conflict of interest.

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
