# Peer review of "Performance Evaluation of Artificial Recharge–Water Intake System Using 3D Numerical Modeling"

_water, doi:10.3390/w14121974_

Round 1
Reviewer 1 Report
The article has well structured and explained the performance of 3-D numerical modeling to evaluate the performance of the artificial recharge-water intake system installed to secure agricultural water in drought areas. However, there are a few minor issues in the English language that need to be improved (i.e., the first paragraph of the conclusion). Secondly, it would be interesting to present the performance of the applied model in case of flash floods, if applicable.
Reviewer 2 Report
Review of the paper
The paper Performance Evaluation of Artificial Recharge-Water Intake 2 System using 3-D Detailed Numerical Modeling refers to study of artificial recharge using 3-D numerical modeling. The artificial recharge is the important solution for water intake in case of shortage of natural recharge. Due to 3-D numerical modeling the optimal design factors were derived.
However, the study are interesting the explanation of research is not sufficient.
The study Area.
The description of study area should be improved.
The lack of information about aquifer, hydrogeological condition, ground water level aquifer etc..
What is the variability of the hydrogeological conditions of the research area? Hydrogeological section or profile of vertical wells are needed.
Methods
This section in not sufficient. Some information should be presented more precisely.
What about precipitation and groundwater exploitation? Has they been included in the model?
line 95 - please explain DEM
line 111 please explain (10/30/50/70/90 mm)
line 113 please explain (5/10/15 mm)
line 115 (10/20/30%)
Results
line 129 " it was divided into 10, 30, 50, 70, 90 129 mm, Please write more details, include the scheme.
Figure 5.
Explain 1,2,3,4
Minor issues
Study Area
line 75 "The geography of the 75 basin is surrounded by mountains...." Please correct the style of this sentence.
Figure 6.
Hydraulic head (m) (above sea level?)
Round 2
Reviewer 2 Report
The article has been improved, but it could have been even better. Please also include the hydrogeological profile or at least the hydrogeological profile of the vertical well
Author Response
Please see the attached file, thanks.
